# Wearable Biofeedback System to Induce Desired Walking Speed in Overground Gait Training

**DOI:** 10.3390/s20144002

**Published:** 2020-07-18

**Authors:** Huanghe Zhang, Yefei Yin, Zhuo Chen, Yufeng Zhang, Ashwini K. Rao, Yi Guo, Damiano Zanotto

**Affiliations:** 1Department of Mechanical Engineering, Stevens Institute of Technology, Hoboken, NJ 07030, USA; hzhang82@stevens.edu (H.Z.); yyin14@stevens.edu (Y.Y.); yzhan21@stevens.edu (Y.Z.); 2Department of Electrical and Computer Engineering, Stevens Institute of Technology, Hoboken, NJ 07030, USA; zchen39@stevens.edu (Z.C.); yguo1@stevens.edu (Y.G.); 3Department of Rehabilitation & Regenerative Medicine, Columbia University, New York, NY 10032, USA; akr7@columbia.edu

**Keywords:** wearable biofeedback system, real-time gait parameter estimation, instrumented footwear, SportSole, closed-loop control

## Abstract

Biofeedback systems have been extensively used in walking exercises for gait improvement. Past research has focused on modulating the wearer’s cadence, gait variability, or symmetry, but none of the previous works has addressed the problem of inducing a desired walking speed in the wearer. In this paper, we present a new, minimally obtrusive wearable biofeedback system (WBS) that uses closed-loop vibrotactile control to elicit desired changes in the wearer’s walking speed, based on the predicted user response to anticipatory and delayed feedback. The performance of the proposed control was compared to conventional open-loop rhythmic vibrotactile stimulation with N = 10 healthy individuals who were asked to complete a set of walking tasks along an oval path. The closed-loop vibrotactile control consistently demonstrated better performance than the open-loop control in inducing desired changes in the wearer’s walking speed, both with constant and with time-varying target walking speeds. Neither open-loop nor closed-loop stimuli affected natural gait significantly, when the target walking speed was set to the individual’s preferred walking speed. Given the importance of walking speed as a summary indicator of health and physical performance, the closed-loop vibrotactile control can pave the way for new technology-enhanced protocols for gait rehabilitation.

## 1. Introduction

Walking speed is a responsive measure of functional status and overall health [1]. Research has shown that decreased walking speed is associated with fear of falling, and increased stride-to-stride (STS) variability in walking speed is a predictor of future falls in the elderly [2]. Additionally, for community-dwelling elderly, low walking speed is associated with mortality [3]; the decline in walking speed is predictive of disability [4]; and lower preferred walking speed was found to be a consistent risk factor for cognitive impairment and institutionalization [5]. Walking speed may also serve as an effective screening parameter for frailty in the elderly [6,7,8] and as an indicator of neurological or musculoskeletal disorders [9,10]. Additionally, many exercise programs to improve fitness, endurance, strength, and balance in the elderly include a walking exercise component as part of broader multi-faceted interventions [11,12,13], and self-selected walking speed is often regarded as an essential outcome of these interventions [14].

Early research on technology-augmented walking exercises has focused on external rhythmic cueing to promote desired changes in gait patterns. Because the cyclical coordination of human gait patterns reflects rhythmical spinal circuits known as central pattern generators (CPG), rhythmic auditory stimulation (RAS) has been used to induce immediate changes in gait patterns by leveraging entrainment effects between a rhythmic auditory signal (typically a metronome beat or a rhythmic music piece) and CPG [15]. Evidence shows that walking speed approximately increases with the square of cadence when cadence is between 80 and 120 steps per minute [16]. Based on this empirical relationship, fixed-tempo rhythmic cues are used in RAS-based protocols to increase walking speed in physically impaired individuals by inducing faster cadence [15,17,18,19]. While RAS has been extensively applied to gait rehabilitation of patients with Parkinson’s disease (PD), traumatic brain injury (TBI), stroke, and cerebral palsy (CP) [20], it has several limitations. First, the relationship between cadence and the square root of velocity is approximate, since velocity is clearly a function of both cadence and stride length [16]. Thus, target improvements in walking speed cannot be reliably predicted by imposing target cadence increments. For instance, McIntosh et al. [18] set the cadence to be 10% faster than the baseline and found an average 14.9% increase in velocity. Lopez and co-workers [19] achieved 38.1% improvement in walking speed by inducing 25% faster cadence. Second, RAS-based protocols assume that people walk at a consistent velocity. This is not true in real-life walking tasks, which involve changes in speed and direction. Third, open-loop rhythmic cues, which are independent of an individual’s own motion, might make him/her not feel engaged in the exercise and stop following RAS, since the external stimuli follow the same fixed rhythmic pattern regardless of the individual’s response.

In the last decade, wearable biofeedback systems (WBS) have been developed to overcome these limitations. A common feature of WBS is the capability of modulating auditory or haptic feedback on-the-fly, based on the wearer’s motor response. Baram and Miller [21] developed a wearable motion and biofeedback sensor to deliver closed-loop auditory cues through headphones and obtained both immediate and short-term improvements in the gait of 14 patients with multiple sclerosis (MS). Hove et al. [22] proposed an interactive RAS method to re-establish natural gait variability in patients with PD. Nagy et al. [23] showed that a rhythmic auditory feedback system could improve gait symmetry in children with spastic hemiparesis during gait training. However, auditory feedback is obtrusive and therefore hardly applicable in real-life public environments, besides being unsuitable for patients with hearing impairments or auditory processing disorders. More recently, vibrotactile feedback systems have been proposed as alternatives to auditory feedback systems for gait rehabilitation. Winfree et al. [24] developed a shoe-based vibratory feedback system to prevent freezing of gait in PD patients. Afzal et al. [25] developed a wearable system to provide real-time vibrotactile cueing to augment gait symmetry training. Yasuda et al. [26] proposed a vibratory cueing system to increase walking speed in patients who suffered a stroke. The system progressively increases the frequency of the cues during a walking session, following a predefined, empirically-tuned Weber law [27].

Besides being applied to patients with neurological disorders, vibrotactile feedback systems have also been used for gait training in amputees. Sharma et al. [28] used eccentric rotating mass (ERM) vibration motors to generate artificial sensory feedback to improve the mobility of lower-limb amputees (LLA). Crea et al. [29] developed a wearable system featuring a pressure-sensitive insole under the prosthetic foot and a set of ERM motors placed on the thigh skin. This system was tested with three elderly LLA, and significant improvements were found in temporal gait symmetry immediately after training [30]. Lauretti et al. [31] proposed a vibrotactile feedback system for improving postural control functions in LLA through restoring plantar pressure perception. Vibrotactile-based feedback systems have also been used to enhance weight shifting stability in LLA, with the goal of improving their balance and reducing the risk of falling [32]. Despite the growing body of research on WBS for gait training, none of the systems and methods developed to date have addressed the problem of eliciting a desired walking speed in the wearer. Because gait speed is a common outcome of walking programs and other exercise-based therapies, this capability would pave the way for new technology-enhanced interventions.

Past research has evidenced that on-line manipulation of auditory reafferences (i.e., the sounds created by a person’s own movements) through a WBS can affect locomotion in a predictable fashion. The bidirectional link of real-time feedback (i.e., perception) and movement execution (i.e., action) can be described in the theory of action-perception coupling [33]. According to this theory, action and perception share common mechanisms and are functionally equivalent. Movement is controlled by comparing expected feedback (generated by internal models) and actual feedback (i.e., reafference) resulting from a motor command [34], and this comparison leads to movement calibration. Thus, action execution may be affected by artificially manipulating sensory reafferences. For instance, Menzer et al. [35] demonstrated that individuals wearing a WBS adjusted their overground walking speed following a sinusoidal function of the time-delay with which auditory cues corresponding to their own footfalls were delivered. Kannape et al. [36] reported subconscious, systematic sinusoidal modulations in stride time in treadmill walking, when controlled audiovisual delays were delivered between a person’s own moments and the representation of such movements by an avatar. Kennel et al. [34] found similar results in hurdling tasks, wherein delayed auditory feedback resulted in increased time to complete the task. Taken together, these studies provide evidence that controlled anticipatory and delayed feedback induces automatic modulations in individuals’ cadence, which lead to systematic adjustments in walking speed.

The contributions of this work are: (i) a new, minimally obtrusive WBS and (ii) a new closed-loop vibrotactile stimulation method to induce a desired walking speed in the wearer during overground walking tasks. The WBS, built upon our previous work on instrumented footwear [37,38,39,40,41,42], is capable of measuring the stride velocity and phase of the gait cycle in real-time during overground walking tasks. These are used as inputs to a closed-loop biofeedback engine that leverages the effects of sensory reafferences’ modulation to elicit desired changes in the wearer’s gait velocity. This is achieved through the modulation of the phase offset at which discrete plantar vibrotactile stimuli are delivered to the wearer. To demonstrate the proof-of-concept feasibility of the new closed-loop stimulation method, we conducted tests with a group of healthy adults who were instructed to walk along an oval path using the WBS under two plantar stimulation conditions: (i) open-loop vibrotactile stimuli (i.e., the conventional method based on fixed-tempo vibrotactile stimulation akin to RAS) and (ii) the proposed closed-loop vibrotactile control. The goals of these tests were to determine whether the proposed closed-loop control strategy: (i) preserves the natural variability of a person’s gait and (ii) results in lower velocity errors compared to the conventional open-loop stimulation, under two experimental conditions (constant and time-varying target gait speed).

The rest of the paper is organized as follows. The architecture of the WBS is described in Section 2. Section 3 describes the closed-loop vibrotactile control. The experimental protocol is illustrated in Section 4. Data analysis is described in Section 5. Results are presented in Section 6 and discussed in Section 7. Finally, the paper is concluded in Section 8.

## 2. Mechatronic Design

The WBS (Figure 1) consisted of two insole modules, a data logger, and a vibration-control unit. Plantar pressure and foot kinematic data were measured at 500 Hz by a multi-cell piezo-resistive sensor and by an inertial measurement unit (IMU), both embedded in the insole. The sensors (IEE S.A., Luxemburg) were located underneath the calcaneus, the lateral arch, the head of the first, third, and fifth metatarsals, the hallux, and the toes. The IMU (Yost Labs Inc., Portsmouth, OH, USA) was placed along the midline of the foot. Data from those sensors were collected by a microcontroller (32-bit ARM Cortex-M4) powered by a small 400 mAh Li-Po battery, both housed in a custom plastic enclosure that was secured to the postero-lateral side of the user’s shoes with a plastic clip. These data were processed on-board and streamed to a remote battery-powered single-board computer (UDP over a local IEEE 802.11n WLAN, 300 Mbps wireless data rate) running the data logger software; see Figure 2.

Compared with the previous version of the device [43], the insole of the WBS was redesigned to house four ERM motors (diameter 10 mm, resonant frequency 240 Hz), in addition to the IMU and multi-cell piezo-resistive sensor. Two ERM motors were located underneath the calcaneous and two underneath and between the hallux and long toe, where the density of the cutaneous mechanoreceptors in the foot sole was higher [44]. The number of ERM motors was selected based on previous work by our group [45], as well as preliminary tests with the WBS.

The vibration-control unit included a control board featuring two transistors (TIP 120) to activate the ERM motors and a power source (3.7 V, 1 Ah Li-Po battery). The vibration-control unit was connected to the logic board of the WBS, which ran the biofeedback engine. The total weight of the WBS was less than 120 g.

## 3. Biofeedback Control

### 3.1. Control Architecture

As shown in Figure 3, the proposed WBS control was composed of three modules: real-time velocity estimator, gait phase estimator, and PI controller. The real-time velocity estimator (described in Section 3.2 below) was based on double integration of foot accelerometry data corrected with de-drifting techniques [46,47]. The gait phase estimator (described in Section 3.3) determined the current gait phase $\phi_{h}$ using adaptive oscillators, gait event detection, and a phase error compensator. The PI controller (described in Section 3.4) took the speed error as the input variable and generated the target phase difference $\Delta \phi_{d}$, which was then added to $\phi_{h}$ to obtain the target phase $\phi_{WBS}$. The latter was fed into the stimulation engine to generate the plantar stimuli.

### 3.2. Real-Time Velocity Estimator

The estimation of real-time stride velocity ($SV_{h}$) started from the determination of gait events, i.e., the timing of initial contact (IC), foot-flat (FF), and toe-off (TO) events, based on the underfoot multi-cell piezo-resistive sensor [48]. After compensating for gravity, zero velocity update (ZUPT) [46] was implemented in the first integration of the acceleration to compute the velocity of the foot over time. The goal of ZUPT was to reinitialize the velocity of the foot at each FF event. Despite gravity removal, the integration of raw accelerometry data accumulated measurement errors, causing drift in the results. To correct the sensor drift, velocity drift compensation (VDC) [47] was adopted in the second integration to obtain the drift-free position of the foot over time. After integrating the dedrifted velocity signal, the real-time STS estimates of stride length ($SL_{h}$) were calculated as the L2-norm between the first and last position of each stride. Stride velocity ($SV_{h}$) was defined as the ratio between the $SL_{h}$ and the corresponding stride time ($ST_{h}$). $ST_{h}$ was computed as the time interval between two consecutive ICs of the same foot.

### 3.3. Gait Phase Estimator

A pool of *M* adaptive frequency oscillators (AFO) was adopted as the gait phase estimator [49,50]. It took the measured foot pitch angle $\theta_{p}$ (i.e., the sagittal-plane angle between the foot sole and the ground) as the input and compared it with the estimated angle $\tilde{\theta }$, written as: (1)$$\tilde{\theta }\left(t\right)=\theta_{0}+\sum_{i=1}^{M}\alpha_{i}sin\left(\phi_{i}\left(t\right)\right)$$
to update the offset $\theta_{0}$, amplitudes $\alpha_{i}$, and phases $\phi_{i}$. The estimated gait phase was determined by the phase of the dominant harmonic $\phi_{1}$. In addition, the phase error compensator initially proposed by Yan et al. [49] was implemented to ensure the null phase at each foot contact. The corrected phase $\phi_{h}$ is computed as: (2)$$\phi_{h}\left(t\right)=mod\left[\phi_{1}\left(t\right)-\phi_{e}\left(t\right),2\pi \right]$$
where $\phi_{e}$ is a smooth phase correction term, being updated based on the difference between the estimated null phase and the actual timing of the IC detected by the multi-cell piezo-resistive sensor.

### 3.4. PI Controller and Stimulation Engine

The stimulation engine prompted the user to adjust his/her walking speed by transforming $\phi_{WBS}$ into a rhythmic vibrotactile cue for IC. The four ERM motors were driven by a sequence of square pulses of width 200 ms where each pulse was initiated whenever $\phi_{WBS}$ crossed zero: (3)$$\phi_{WBS}\left(t\right)=mod\left[\phi_{h}\left(t\right)+\Delta \phi_{d},2\pi \right]$$
The phase-shift of the vibrotactile stimuli relative to the IC ($\Delta \phi_{d}$) was automatically adjusted by the PI controller. Specifically, if the user’s current walking speed ($SV_{h}$) was slower than the target speed ($SV_{T}$) set by the experimenter, the stimuli anticipated the IC to encourage a faster pace. Conversely, if $SV_{h}$ was faster than $SV_{T}$, the stimuli lagged the IC, to elicit a slower pace (Figure 4). The PI gains were manually tuned by the experimenters during preliminary tests preceding the experimental validation.

## 4. Experimental Protocol

N = 10 healthy male adults (age 27.6±1.3 years, height 1.75±0.06 m, weight 75.4±11.5 kg) volunteered for this study, which required a single visit to the laboratory. All participants were healthy adults with no musculoskeletal or neurological problems that would affect their ability to walk for 10 min. The experimental protocol was approved by the Institutional Review Board of Stevens Institute of Technology.

After setting up the WBS, participants were instructed to complete a two minute baseline (BL) test (Figure 5a). During this test, participants were asked to walk counter-clockwise (CCW), along a prescribed oval path marked on the floor (Figure 5b), at a comfortable pace. The last minute of the baseline (BL) test was used to compute the participant’s preferred cadence ($CAD_{P}$), stride velocity ($SV_{P}$), and coefficient of variation (CV) in cadence and stride velocity. Afterwards, participants were asked to complete a familiarization session (FS), which lasted for approximately ten minutes. During this session, participants experienced both open-loop vibrotactile stimuli (OS) and closed-loop vibrotactile stimuli (CS), but were blind to the type of stimuli. The stimuli in OS mode were triggered at a constant pace corresponding to a target cadence $CAD_{T}$, while the stimuli in the CS mode were modulated by the PI controller, given a target velocity $SV_{T}$. The goal of the FS was to help users get accustomed to the WBS; therefore, the values of $CAD_{T}$ and $SV_{T}$ for this session were arbitrarily set by the experimenter. Participants were asked to adjust their gait to the stimuli, but were blinded to the type of rhythmic stimuli and to the purpose of the stimuli. After the FS, participants were instructed to complete a four minute walking bout under each of the two stimulation methods (Session 1, S1). During S1, the target walking speed ($SV_{T}$) for CS and the target cadence ($CAD_{T}$) for OS were set to $SV_{P}$ and $CAD_{P}$, respectively:(4)$$\begin{array}{c} SV_{T}=SV_{P}\\ CAD_{T}=CAD_{P} \end{array}$$
S1 was used to determine whether the two stimulation methods preserved the natural variability of a person’s gait. After S1, participants rested for approximately ten minutes. Session 2 (S2) followed a similar protocol as S1, and $SV_{T}$ for CS was set to: (5)$$SV_{T}=1.15SV_{P}$$
Based on [16], the corresponding $CAD_{T}$ for OS was set to: (6)$$CAD_{T}=CAD_{P}*\sqrt{1.15}$$
S2 was included in the protocol to determine the relative performance of the two types of stimulation during fast walking tasks. In Session 3 (S3), participants were instructed to complete two nine minute walking tasks. During these two tasks, the $SV_{T}$ for CS was set to oscillate between $0.85SV_{P}$ and $1.15SV_{P}$, within a period of 240 s: (7)$$SV_{T}\left(t\right)=SV_{P}-0.15SV_{P}\sin\left(2\pi \frac{1}{240}t\right)$$
The corresponding $CAD_{T}$ for OS was calculated as: (8)$$CAD_{T}\left(t\right)=CAD_{P}*\sqrt{SV_{T}\left(t\right)/SV_{P}}$$
The sequence of OS and CS in all sessions was randomized. For all the walking tasks, participants were instructed to walk CCW, along a prescribed oval path marked on the floor.

## 5. Data Analysis

For S1 and S2, only the gait cycles occurring in the last minute of each test were included in the data analysis. For S3, the last four minutes of each test (i.e., one full period of the target speed function described by (Equation 7)) were included in the data analysis. It took participants less than two minutes to have their gait patterns adapted to the OS or CS stimuli, and the analyzed data did not include the data acquired during the time of the gait adaptation process. For S1, the average values of SV and CAD and their coefficients of variation were chosen as the outcome metrics. For S2 and S3, the percentage mean absolute error (MAE%) of SV and CAD (computed with respect to their target values $SV_{T}$ and $CAD_{T}$) and the coefficient of variation of SV and CAD were selected as the outcome metrics.

One sample Wilcoxon signed-rank tests were used to identify significant (α<0.05) effects of the two stimulation methods on participant’s natural gait during S1. This was achieved by computing the average relative changes of SV and CAD (and their coefficient of variation) with respect to the BL values and then comparing these to the null value. Paired Wilcoxon signed-rank tests were conducted to assess significant effects of the two stimulation methods on gait adaptation performance under two experimental conditions (i.e., constant and time-varying target gait speed, corresponding to S2 and S3, respectively). This was done by comparing the MAE% of SV and CAD and their CV corresponding to OS and CS. The Bonferroni–Holm method was applied to correct for the family-wise error rate. All statistical analysis was carried out in SPSS v24 (IBM Corporation, Armonk, NY, USA).

## 6. Results

For the baseline test, participants’ $SV_{P}$ ranged from 96.26 to 146.15 cm/s (117.18 ± 13.36 cm/s, mean and SD), and $CAD_{P}$ ranged from 86.88 to 109.41 steps/min (100.06 ± 6.60 steps/min). Figure 6 shows the group averages (AVG) and the mean CV of the real-time estimates of SV and CAD during S1 compared to their BL values. Figure 7a,b shows percentage changes in SV and CAD as functions of the baseline SV and CAD. Figure 7c,d illustrates the changes in gait variability induced by the two stimulation methods compared to BL. One sample Wilcoxon signed-rank tests revealed no significant differences in SV, CAD, and their coefficients of variation relative to the BL values. Hence, neither OS, nor CS significantly affected the wearer’s natural gait, when the target walking speed or cadence were set to their preferred walking values. Besides, the analysis evidenced no noticeable correlations between the baseline SV and CAD values and the corresponding percentage changes during S1.

Figure 8a,b illustrates the MAE% and the standard error (SE) of SV and CAD with respect to the target velocity and target cadence under the two stimulation methods in S2. Paired Wilcoxon signed-rank tests revealed significantly smaller errors in SV for CS compared to OS (p=0.002); however, the opposite was found for the errors in CAD (p=0.002). Figure 8c,d shows the CV of the estimates of SV and CAD for OS and CS in S2. Paired Wilcoxon signed-rank tests indicated no significant differences between OS and CS in terms of CV, both for SV and for CAD. Figure 9a,b and Figure 10a,b show the real-time cadence ($SV_{h}$) and stride velocity ($CAD_{h}$) of a representative participant under the two stimulation methods in S3. Figure 9c and Figure 10c illustrate the MAE% of SV and CAD for OS and CS stimulation in S3. Similar to what was found in S2, paired Wilcoxon signed-rank tests revealed significantly smaller velocity errors (p=0.0137), but larger cadence errors (p=0.002) for CS compared to OS.

To summarize, CS resulted in lower velocity errors than OS, both when the target speed was a constant term and when it was defined as a time-varying function. Conversely, OS could effectively induce adjustments in cadence, both at fixed and at time-varying target velocity, but these adjustments did not correspond to the modulations in gait velocity predicted by the approximate relationship between CAD and SV, which is implicitly assumed by conventional RAS protocols. Besides, the two stimulation methods performed similarly in terms of gait variability. Additionally, no noticeable correlation was found between MAE% in SV and the participants’ baseline velocity.

## 7. Discussion

This paper proposed a new, minimally obtrusive WBS to elicit desired changes in the wearer’s walking speed during overground walking exercises. The WBS measured stride-to-stride velocity and adjusted on-the-fly the phase shift at which plantar vibrotactile stimuli were provided at each step. These stimuli, in turn, were expected to modulate sensory reafferences in the wearer, in order to influence their movement control towards the target walking speed. While biofeedback systems have been extensively used in walking exercises for gait improvement, there is a paucity of literature addressing the problem of controlling gait speed. To the best of the authors’ knowledge, this was the first study to demonstrate the use of a biofeedback system to control the wearer’s walking speed during overground walking tasks. As self-selected gait speed is a common outcome measure in exercise-based therapies that aim to improve gait, balance, endurance, and strength, the proposed WBS, which is fully portable, can potentially be used for self-administered, home-based gait exercises for older adults and for patients with neurological disorders.

### 7.1. Open-Loop vs. Closed-Loop Rhythmic Stimulation

While RAS and other protocols based on OS aim to increase walking speed by inducing faster cadence, the approximate relationship between CAD and SV makes it impossible to set the value of $CAD_{T}$ a priori for a desired improvement in gait speed, as indicated by this study. For this reason, past studies on rhythmic stimulation for walking exercises have adopted standard increments, ranging from 10% to 25% of an individual’s preferred pace [18,19,51], which do not directly relate to a target functional outcome. Additionally, RAS protocols are typically limited to straight-line walking over a short distance (e.g., an eight meter walkway [18,19]), which does not resemble real-life walking tasks. In recent years, CS has been proposed to overcome these limitations. The simplest closed-loop stimulation method consists of providing stimuli at specific gait events, such as IC and TO [52]. This method elicited significant improvement in the walking speed of patients with PD, but the improvement was relatively small (i.e., <10%). Besides providing a simple rhythmic beat in response to the patient’s gait events, another method is to produce a continuous rhythmic music piece synchronized with the patient’s gait pattern, rewarding the patient for making an effort to increase walking speed [21,53,54,55,56]. However, this method can hardly elicit a desired walking speed in the wearer. Yasuda et al. [26] applied Weber’s law to progressively decrease the time interval between consecutive stimuli in patients with stroke. With this method, the patients’ cadence was progressively increased, and as a result, their walking speeds also increased. Nonetheless, this method shares the same shortcomings of the OS-based methods.

### 7.2. Proposed Closed-Loop Vibrotactile Control vs. Existing Methods

The closed-loop stimulation method described in this study builds upon previous research, which found that on-line manipulation of auditory reafferences may modify the wearer’s walking speed [35]. Similar to OS, the proposed CS increases walking speed by modulating the wearer’s cadence. Unlike OS, the proposed stimulation method controls anticipatory and delayed feedback to induce automatic modulations in the wearers’ cadence, which leads to systematic adjustments in their walking speed. Experiment testings presented in this paper proved our hypotheses that the proposed CS not only preserves the natural variability of a person’s gait, but also results in better performance in terms of gait speed adaptations compared to OS.

Our results are in line with previous studies on rhythmic stimulation, which found no change in SV [51,57] or CAD [51] of healthy adults with $CAD_{T}$ matched to $CAD_{P}$. However, OS was observed to improve SV in patients with PD [51], even when the stimuli were provided at the individual’s preferred cadence. Similar results were also found with CS methods: improvements in SV were reported in patients with MS [21], cerebral palsy [53], and PD [54], but not in healthy individuals [21,53]. This suggests that both OS and CS may be more suitable for patients with neurological disorders than healthy adults. Because increased gait variability is associated with increased risk of falling [2], the evaluation of potential treatments such as OS and CS should include not only their effects on STS gait parameters, but also on gait variability. Similar to previous studies [57,58], we found that OS did not significantly affect the gait variability of healthy adults. On the other hand, Hausdorff et al. [51] observed reduced STS variability of CAD in patients with PD under OS. Because healthy adults do not have cognitive impairments, they might not benefit from entrainment effects between external cues and CPG. Instead, theses cues might disrupt the normal gait pattern, e.g., through increased attentional demand due to motor adjustments having to be made to follow the cue [57,59]. Conversely, patients with PD seem to rely more on external inputs to guide movement, and thereby, they may benefit from external cues.

CS resulted in better gait adaptations than OS both with constant and with time-varying target walking speeds. This result suggests participants can follow different target walking speeds in one single test, thereby making it possible to simplify training protocols instead of separating preferred pace walking and fast pace walking in multiple tests [18,51]. Due to inter-individual variability, a fixed $SV_{T}$ may suit most, but not all wearers [21]. Because the proposed CS worked for a variety of $SV_{T}$ (i.e., 85–115% $SV_{P}$), it has the potential to avoid falling out of sync in wearers by changing $SV_{T}$ on-line. Results also indicated that the two stimulation methods, OS and CS, performed similarly in terms of gait variability at fast walking speed. However, it was not possible to compare gait variability measured in S2 and S3 with their baseline values, because gait variability is affected by walking speed [60].

### 7.3. Limitations and Future Work

While this study provided promising preliminary results, further research must be conducted in order to assess the feasibility of using on-line modulated plantar stimuli to induce desired changes in the user’s walking speed. First, the relatively small and homogeneous sample size may limit the generalizability of these findings. The proposed CS might not yield the same performance in a more heterogeneous sample or when applied to clinical populations. Therefore, future work will test the proposed CS in older adults, including healthy individuals and patients with movement disorders of both sexes. Second, we assumed all the participants could adapt to the target walking speed. However, this assumption might not hold for the elderly or for patients with disorders affecting the walking function. Thus, future studies must address methods to adapt $SV_{T}$ on-the-fly, based on the user’s response. Third, in this paper, we only addressed immediate changes to cadence and walking speed that occurred through training. Past research [52] reported no functional carry-over effects with OS in patients with PD. However, CS was found to elicit both short-term [51,54] and long-term [55] improvement in walking speed. Hence, future studies will need to assess carry-over effects of training with the proposed CS.

## 8. Conclusions

This paper presented the development and validation of a new and minimally obtrusive WBS that is capable of measuring the walking speed and phase of the gait cycle in real-time during overground walking tasks. A closed-loop vibrotactile stimulation method was proposed to induce desired changes in the wearer’s walking speed based on the user’s response to anticipatory and delayed feedback. The results indicated that the proposed closed-loop vibrotactile stimulation method could induce more accurate gait speed adaptations compared to conventional fixed-tempo (open-loop) stimulation and also preserve the natural variability of a person’s gait. These findings advance the research on wearable biofeedback systems for gait exercises and provide potential solutions for future technology-enhanced exercise-based interventions for patients with gait disorders.

## Figures and Tables

**Figure 1 sensors-20-04002-f001:**
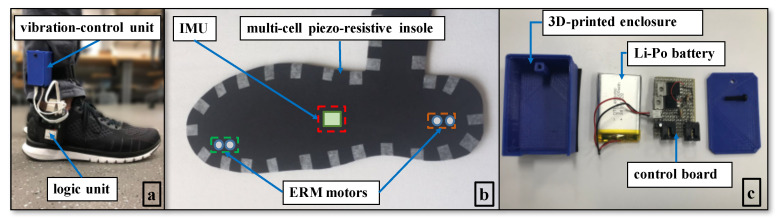
(**a**) The proposed wearable biofeedback system (WBS) consists of two insole modules, a data logger, and a vibration-control unit; (**b**) The insole module includes a multi-cell piezo-resistive sensor, an IMU, and four eccentric rotating mass (ERM) motors, all embedded in the insole; (**c**) The vibration-control unit includes a control board and a Li-Po battery. The instrumented insoles are fitted inside regular sneakers; the logic unit is housed inside a customized 3D-printed enclosure and attached to the wearer’s shoes with a clip; and the vibration-control unit is attached to the user’s distal shank through Velcro straps.

**Figure 2 sensors-20-04002-f002:**
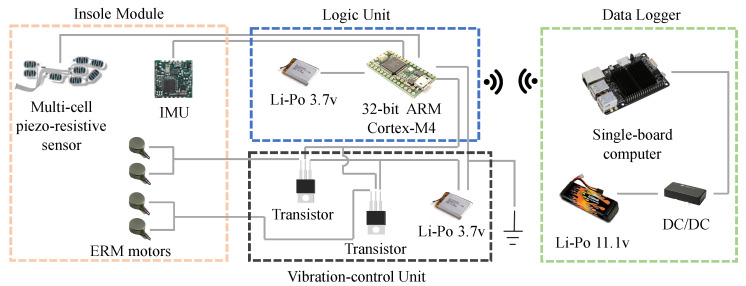
System architecture.

**Figure 3 sensors-20-04002-f003:**
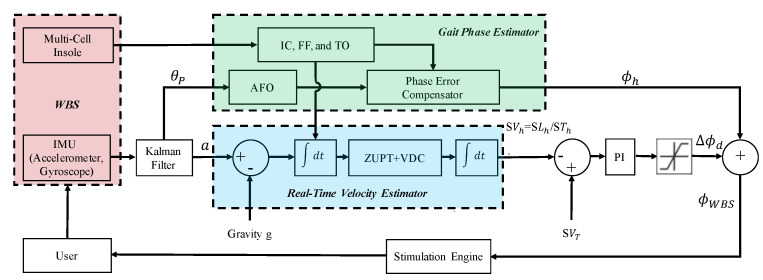
Flowchart of closed-loop vibrotactile control. $\theta_{p}$ is the measured foot pitch angle, and *a* is the foot acceleration. $SV_{h}$, $SL_{h}$, and $ST_{h}$ are the real-time stride-to-stride estimates of stride velocity, stride length, and stride time, respectively. $SV_{T}$ represents the target walking speed. $\phi_{h}$ and $\phi_{WBS}$ are the user’s current gait phase and target phase, respectively. $\Delta \phi_{d}$ is the target phase difference. IC, initial contact; FF, foot-flat; TO, toe-off; AFO, adaptive frequency oscillator.

**Figure 4 sensors-20-04002-f004:**
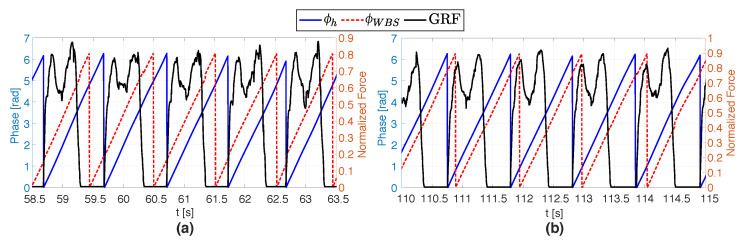
Effects of the closed-loop stimulation on the gait of a representative participant. The black line represents the normalized ground reaction force (GRF) extracted from the multi-cell piezo-resistive sensor. If the user’s current walking speed ($SV_{h}$) is slower than the target speed ($SV_{T}$) set by the experimenter, the stimuli anticipate the IC to encourage a faster pace (**a**); Conversely, if $SV_{h}$ is faster than $SV_{T}$, the stimuli lag the IC, to elicit a slower pace (**b**).

**Figure 5 sensors-20-04002-f005:**
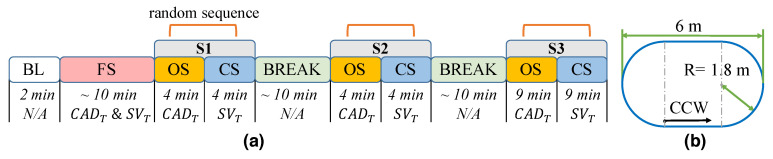
(**a**) Experimental protocol. The sequence of the two rhythmic stimuli (OS = open-loop vibrotactile stimuli, CS = closed-loop vibrotactile stimuli) was randomized. The stimuli in OS mode were triggered at a constant pace corresponding to a target cadence $CAD_{T}$, while the stimuli in CS mode were modulated by the PI controller, given a target velocity $SV_{T}$; (**b**) For all the tasks, participants were instructed to walk counter-clockwise along a prescribed oval path marked on the floor. BL, baseline.

**Figure 6 sensors-20-04002-f006:**
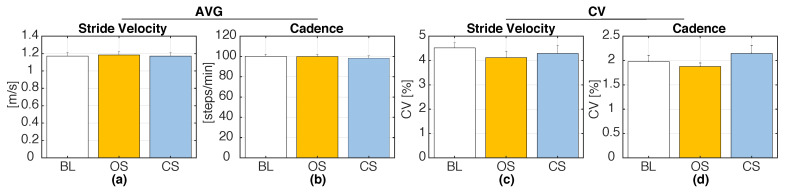
(**a**,**b**) Group averages (AVG) and (**c**,**d**) coefficient of variation (CV) of SV and CAD induced by the two stimulation modes (OS = open-loop vibrotactile stimuli, CS = closed-loop vibrotactile stimuli) during Session 1 (S1), as compared to their baseline (BL) values. Error bars indicate ± 1SE.

**Figure 7 sensors-20-04002-f007:**
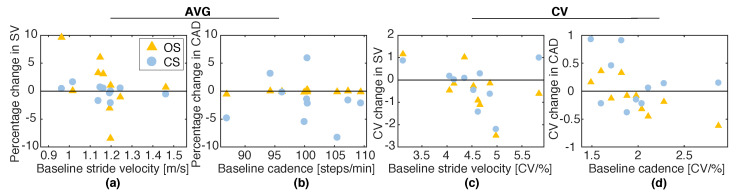
(**a**,**b**) Percentage changes in SV and CAD (relative to their baseline values $SV_{P}$ and $CAD_{P}$) induced by the two stimulation modes during S1, as functions of the baseline values; (**c**,**d**) Changes in the coefficients of variation (CV) of SV and CAD induced by the two stimulation methods during S1, as functions of the baseline CV. In all plots, each mark represents a participant.

**Figure 8 sensors-20-04002-f008:**
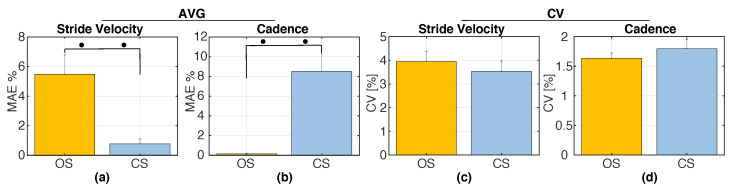
(**a**,**b**) Group averages of the percentage mean absolute errors (MAE%) of SV and CAD induced by the two stimulation methods, OS and CS, during S2. MAE% values are computed with respect to the target values $SV_{T}$ and $CAD_{T}$; (**c**,**d**) Coefficient of variation (CV) of SV and CAD induced by the two stimulation methods during S2. Error bars indicate ± 1SE. •• indicates p<0.01.

**Figure 9 sensors-20-04002-f009:**
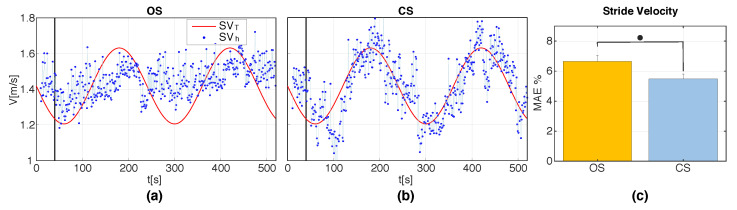
(**a**,**b**) $SV_{h}$ for a representative participant for the two stimulation methods, OS and CS, during S3. The black vertical line represents the time at which the stimulation engine was activated during the walking task; (**c**) Group averages of the percentage mean absolute errors (MAE%) of SV induced by OS and CS during S3. MAE% values are computed with respect to the time-varying target values $SV_{T}\left(t\right)$. Error bars indicate ± 1SE. • indicates p<0.05.

**Figure 10 sensors-20-04002-f010:**
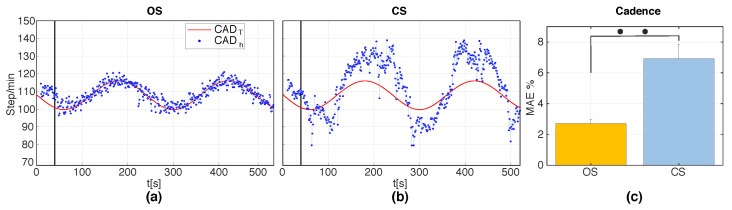
(**a**,**b**) $CAD_{h}$ for a representative participant for the two stimulation methods, OS and CS, during S3. The black vertical line represents the time at which the stimulation engine was activated; (**c**) Group averages of the percentage mean absolute errors (MAE%) of CAD induced by OS and CS during S3. MAE% values are computed with respect to the time-varying target values $CAD_{T}\left(t\right)$. Error bars indicate ± 1SE. •• indicates p<0.01.

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
