# Peer review of "Wearable Biofeedback System to Induce Desired Walking Speed in Overground Gait Training"

_sensors, 2020, doi:10.3390/s20144002_

Round 1

Reviewer 1 Report

This study examines a new wearable biofeedback system (WBS) able to assess gait and provide real-time vibrotactile stimulation (by a closed-loop method) to improve walking speed. To this aim, the authors tested 10 healthy subjects (HS) while performing a walking task and compared the proposed closed-loop vibrotactile stimulation (CS) with a conventional open-loop vibrotactile stimulation (OS). Although the two stimulation protocols showed similar effects on gait variability, the proposed WBF with CS was more accurate than the WBF with OS for improving walking speed. Accordingly, the authors concluded that the proposed WBS with CS may help to enhance locomotion in patients suffering from gait disorders.

The study is certainly of interest. I have, however, several concerns: 

  • The introduction is rather confusing. The authors should reorganize this section by improving the flow of concepts, merging paragraphs 1.1 and 1.2 and reducing the word count. Also, the description of previous findings in the introduction is rather poor and based on a list of studies. Accordingly, these data should be better summarized in order to explain the rationale of the study.  
  • Changes in walking speed are compensatory mechanisms in patients with gait or balance disorders rather than reflecting independent factors leading to functional gait abnormalities (e.g. increased stride-to-stride variability, higher risk and fear of falling) (Page 1, Lines 20-28). It is therefore questionable that enhancing velocity would lead to the improvement of gait. This point is quite unclear and should be discussed in the introduction and discussion sections more in detail. 
  • The authors should clarify the meaning of open-loop rhythmic cues when first referring to this concept in the introduction (Page 2, Lines 46-48). 
  • Spatio-temporal parameters of gait are strictly influenced by walking speed. It is not clear why, besides cadence, the authors considered speed more convenient than other gait parameters for real-time biofeedback.
  • The paragraph 3 (Page 4, lines 131-144) mainly reports background data rather than methods. Accordingly, this paragraph should be included in the introduction section.
  • The paragraph 5.1 (Page 7, lines 185-199) should be moved in the methods section since it includes methodological issues. 
  • The authors should clarify the term "functional gait" (e.g. Page 2, Line 45; Page 10, Line 248).
  • The paragraph 6.1 includes several background data already reported in the introduction and does not discuss sufficiently the main results of the study. Accordingly, this paragraph should be improved by better speculating about the new findings.      

Author Response

Please see the response in the attached document.

Reviewer 2 Report

This is a very comprehensive article, and suitable for publication in SENSORS journal:

- The system is based on sensors and actuators.

- The application is useful.

- The system is novel.

- The results are extensive and promising.

- A very complete discussion of the benefits of the system has been carried out, comparing with results from other authors.

For an improvement in the quality of work, the reviewer recommends making a detailed description of the hardware and components used in the system. It is recommended to include a figure with a schematic diagram of the main components and systems. A detailed diagram / schematic explanation would be recommended to explain the functionality of the different components, detailing the model and manufacturer of the main elements.

Author Response

(The authors gave the same response as above.)

Reviewer 3 Report

Please see the comments in the attached document.

Author Response

(The authors gave the same response as above.)

Reviewer 4 Report

In this article the authors have proposed a new method to provide vibrotactile stimulation. Closed-loop control was developed and compared with an open-loop by measuring cadence, stride velocity, and coefficient of variation. The manuscript is well written and conveys a clear message. However, some minor revisions could improve the readability.

Authors should clarify how they imagine the applicability of the proposed approach in everyday life.

How should closed-loop vibrotactile control be handled in the functional gait, where the speed of gait is often adjusted according to environments and tasks?

What is the delay in the subject's response to closed-loop vibrotactile control?

In experimental sessions S1, S2, and S3, the authors provide 4 minutes of closed-loop and open-loop activity without interruption. Since rhythmic stimuli have been shown to provide effects on subsequent trials, how did the authors manage the carry-over effect in the experimental session?

From raw 184 to 199, the authors report the data and the statistical analysis. This section should be moved into the materials and methods section.

Author Response

(The authors gave the same response as above.)

Round 2

Reviewer 1 Report

The authors have satisfactorily addressed my comments. 

Reviewer 3 Report

Please see my recommendation in the attached document.
